# Carbohydrates, enzyme activities, and microbial communities across depth gradients in the western North Atlantic Ocean

5 C. Chad Lloyd<sup>1</sup>, Sarah Brown<sup>2</sup>, Greta Giljan<sup>3</sup>, Sherif Ghobrial<sup>1</sup>, Silvia Vidal-Melgosa<sup>3,4</sup>, Nicola Steinke<sup>3</sup>, Jan-Hendrik Hehemann<sup>3,4</sup>, Rudolf Amann<sup>3</sup>, Carol Arnosti<sup>3,1</sup>

Correspondence to: C. Arnosti (carnosti@mpi-bremen.de)

Abstract. Heterotrophic bacteria process nearly half of the organic matter produced by phytoplankton in the surface ocean. Much of this organic matter consists of high molecular weight (HMW) biopolymers such as polysaccharides and proteins, which must initially be hydrolyzed to smaller sizes by structurally specific extracellular enzymes. Few previous studies, however, have investigated the structural complexity of polysaccharides among regions and depths. To simultaneously investigate substrate structure and microbial community composition and function, we concurrently determined carbohydrate abundance and structural complexity, bacterial community composition, and peptidase and polysaccharide hydrolase activities across depth gradients from surface to bottom water at four distinct stations in the western North Atlantic Ocean. Although the monosaccharide constituents of particulate organic matter (POM) were similar among stations, the structural complexity of POM-derived polysaccharides varied by depth and station, as demonstrated by polysaccharide-specific antibody probing. Bacterial community composition and polysaccharide hydrolase activities also varied substantially by depth, suggesting that the structure and function of bacterial communities may be related to substrate structural complexity. Thus, the extent to which bacteria can transform organic matter in the ocean is dependent on both the structural complexity of the organic matter and their enzymatic capabilities in different depths and regions of the ocean.

25

**Short Summary.** Carbon cycling throughout the ocean is dependent on the balance between phytoplankton productivity and heterotrophic decomposition. Bacteria must produce structurally specific enzymes to degrade specific chemical structures found in organic matter. Organic matter composition, environmental physical/chemical parameters, microbial community composition, and enzymatic activities varied with depth; the structural complexity of organic matter varied also with location in the ocean.

<sup>&</sup>lt;sup>1</sup>Department of Earth, Marine, and Environmental Sciences, University of North Carolina at Chapel Hill, Chapel Hill, USA

<sup>&</sup>lt;sup>2</sup>Environment, Ecology, and Energy Program, University of North Carolina at Chapel Hill, Chapel Hill, USA

<sup>&</sup>lt;sup>3</sup>Department of Molecular Ecology, Max Planck Institute for Marine Microbiology, Bremen, Germany

<sup>&</sup>lt;sup>4</sup>MARUM: Center for Marine Environmental Sciences, University of Bremen, Bremen, Germany

## 1 Introduction

Polysaccharides and proteins are major macromolecular components of the phytoplankton-derived organic matter (Wakeham et al., 1997; Hedges et al., 2002) that forms the base of marine food webs. Much of this organic matter is processed by heterotrophic bacterial communities (Azam and Malfatti, 2007) that use structure-specific extracellular enzymes (Lombard et al., 2014) to hydrolyze high molecular weight (HMW) organic matter to sizes suitable for uptake (Fig 1a). Since different community members possess distinct enzymatic complements (Avci et al., 2020, Krüger et al., 2019), the composition of a microbial community is an important determinant of a community's enzymatic capabilities. These community enzymatic capabilities vary with depth and location in the ocean (Arnosti et al., 2011; Steen et al., 2012; Hoarfrost and Arnosti, 2017; Balmonte et al., 2021). However, the cues to which these microbial communities are responding—in particular, the structure of the macromolecules that they sense and target—have to date not been well characterized.

This long-standing knowledge gap (Hedges et al., 2000) is due in part to the difficulties of characterizing intact macromolecules. These macromolecules are found as part of the high molecular weight dissolved (HMW DOM) as well as the particulate (POM) phases of organic matter. POM is operationally defined by filtration – organic matter retained by a filter of defined pore size – and DOM is the filtrate; large volumes of DOM can be concentrated (often via ultrafiltration) to HMW DOM. The macromolecules in HMW DOM as well as in POM include proteins and polysaccharides, which are typically broken into component pieces for further analysis. Although recent analytical advances have enabled high-resolution identification of protein structures in marine organic matter (e.g. Morris et al., 2010; Saito et al., 2019; Francis et al., 2021; Saunders et al., 2022), comparable analyses are not yet possible for polysaccharides. A major issue is the fact that the individual monosaccharides that make up a polysaccharide can be linked in multiple positions, unlike the amino acids that

Fig. 1: a) Key factors affecting biogeochemical cycling of high molecular weight (HMW) organic matter in the ocean: macromolecular structure, microbial community composition/capabilities, and the activities of the extracellular enzymes that process specific macromolecules. b) Conceptual representation of three different groups of macromolecules (1-3) that could lead to the same pool of hydrolyzed monomers (4). Analysis of the monomers provides no information on the original sequence or 3D structure of the parent macromolecule(s) – structural features that are essential for enzyme-substrate fit.

make up proteins. As a result, even a few monosaccharides can be linked in multiple ways (Laine, 1994). Monomer analysis thus cannot provide information about the order of the monomers in a polymer chain or the 3D structure of the initial macromolecule (Fig. 1b), aspects of structure that are essential for enzyme-substrate 'fit'.

Interrelated factors therefore play a key role in the biogeochemical cycling of organic matter in the ocean: the structure of marine organic matter, the capabilities of the microbial communities that process it, and the activities of the enzymatic tools they use to initiate its degradation (Fig. 1a). Here, we investigate these factors through a variety of approaches. We use recent advances in carbohydrate chemistry (Vidal-Melgosa et al., 2021; Buck-Wiese et al., 2023) that have previously been used only once in the open ocean (Priest et al., 2023) to derive new insight into the structural complexity of marine carbohydrates. Currently, however, these advanced analytical techniques can only be carried out on POM or on HMW DOM. We use well-established methods to determine composition of microbial communities from water samples collected at different depths and locations in the ocean. However, with the exception of 'selfish' bacteria (Cuskin et al., 2015; Reintjes et al., 2017), it is not yet possible to definitively link specific microbial taxa in these communities with the activities of the extracellular enzymes they release to initiate the degradation of HMW organic matter: these enzyme activities currently can only be measured on a whole-community level. We therefore measured community composition and the activities of these communities' enzymes in similar water volumes from the same samples. Given the relatively small volumes of water used in our analyses, the bacterial community – as well as the activities of the enzymes we measured – likely predominately represents the free-living fraction; the carbohydrate analyses, due to the requirement for higher concentrations of organic matter, were carried out on POM.

Through the depth of the water column at four distinct sites in the western North Atlantic Ocean, we measured the activities of enzymes targeting polysaccharides and proteins, microbial community composition, and carbohydrate constituents of organic matter. These stations are characterized by different water masses and levels of primary productivity, and thus are likely to have distinct microbial communities. Although previous investigations have compared bacterial community composition with polysaccharide hydrolase activities (e.g., Murray et al., 2007; Teske et al., 2011; Arnosti et al., 2012; Balmonte et al., 2018; Giljan et al., 2023; Lloyd et al., 2023), this is the first study to also investigate the structure and abundance of the combined carbohydrates these communities may target.

## 2 Methods

85

## 2.1 Stations and water sampling

Water samples were collected in the western North Atlantic aboard R/V *Endeavor* (cruise EN638; May 15th—May 30th, 2019). Samples were collected at one continental slope station and three open ocean stations (Stns. 17 and 18-20, respectively; Fig. 2; Table S1) using a Niskin rosette (12 x 30L bottles) equipped with a Seabird 32 CTD. At each offshore station, water was collected from seven discrete depth horizons including the surface (~ 3 m), deep chlorophyll maximum (DCM; as observed from the CTD's fluorescence sensor), 300 m, the lower mesopelagic (typically between 600 and 850m), 1500 m, 3000 m, and bottom (ranging from 3190 – 5580 m). At Stn. 17, water was collected from the surface, DCM, and bottom (depth: 625 m). Seawater was transferred from Niskin bottles into 20 L carboys that had been acid washed and rinsed with reverse osmosis water, then rinsed three times with seawater from the sampling depth prior to filling. Transfers were carried out using silicone tubing that had been acid washed and rinsed with reverse osmosis water prior to use. Seawater was taken directly from the carboys to measure bacterial protein production, polysaccharide hydrolase activities, peptidase activities, and glucosidase activities, as described below.

Temperature (T), salinity (S),  $O_2$ , and chlorophyll-a concentration data (Table S2) are all from the Seabird CTD sensors. Chlorophyll-a was calculated by multiplying the scaling factor (in  $\mu$ g L<sup>-1</sup> V<sup>-1</sup>) by the sensor corrected sensor output (Output – Dark Counts, in V) to obtain a concentration in  $\mu$ g L<sup>-1</sup>. This value was then converted to mg m<sup>-3</sup> (see Table S2).

## 2.2 Bacterial production and cell counts

## 2.2.1 Bacterial productivity

Bacterial productivity was measured after Kirchman et al. (2001), using incorporation of tritiated leucine (<sup>3</sup>H-Leu; 20 nM). In brief, leucine incorporation rates were measured in samples incubated in the dark at *in-situ* temperatures. Leucine incorporation rates are reported as pmol leucine per L per hour.

#### 2.2.2 Bacterial cell counts

Seawater samples were fixed with formaldehyde at a final concentration of 1%, then 25-50 mL of the fixed samples were filtered through a polycarbonate filter (pore size: 0.22 µm) at a maximum vacuum of 200 mbar. DNA staining was done using 4′,6-Diamidin-2-phenylindol (DAPI), and samples were mounted with a Citifluor/VectaShield (4:1) solution.

Cell counting was done using a fully automated epifluorescence microscope (Zeiss AxioImager.Z2 microscope stand, Carl Zeiss) and image analysis was carried out as described by Bennke et al. (2016). Cell count verification of the automated image analysis was done manually.

## 110 2.3 Organic matter analyses of POM

## 2.3.1 Collection of particulate organic matter

To determine the structural complexity of carbohydrates (epitope analysis; see below), the current state of the art require analysis of particulate organic matter (POM) collected on a glass fiber filter or HMW DOM concentrated via ultrafiltration; the concentration of carbohydrates dissolved in seawater is far too low for direct detection using this technique. Since we did not have access shipboard to equipment that would be required for ultrafiltration, we focused on POM composition. The POM was collected by filtering 5-15 liters of water through a 47-mm pre-combusted (400°C for 6 hours) glass fiber filter (GF/F; nominal pore size 0.7 µm; for volumes filtered at each depth and station, see Table S3). These samples were collected at all depths and stations.

## 2.3.2 Particulate organic carbon

Particulate organic carbon was measured as described in Becker et al. (2020). In brief, triplicate filter punches from samples collected on pre-combusted (400°C for 6 hours) glass fiber filters (GF/F) were placed in an acidic environment (concentrated HCl fumes) for 24 h to remove inorganic carbon. After drying for 24 h at 60 °C, the samples were packed in pre-combusted tin foil. C was quantified using an elemental analyzer (cario MICRO cube; Elementar Analysensysteme) using sulfanilamide for calibration. Limits of detection for POC was 0.001 mg C/L, based on the standard deviation of blank measurements.

#### 2.3.3 Monosaccharide composition of POM

The monosaccharide constituents of total combined carbohydrates (i.e., polysaccharides, glycoproteins, glycolipids, etc.) of POM (collected as described above) were determined from triplicate filter punches (11.2 mm diameter). Samples were acid hydrolyzed by adding 1 M HCl to each filter piece, flame sealing each piece in a glass ampule, and placing the ampules in a drying oven at 100°C for 24 hours. After acid hydrolysis, the samples were dried on a speed-vac and resuspended in Milli-Q to remove any HCl. The quantity and composition of the resulting monosaccharides were measured using a modified protocol (Engel and Handel (2011), as described by Vidal-Melgosa et al. (2021)). In brief, neutral, amino, and acidic sugars were quantified using high performance anion exchange chromatography on a Dionex ICS-5000+ system with pulsed amperometric detection (HPAEC-PAD). Peaks were identified using retention times of purified monosaccharide standards; abundance was quantified from standards using the peak area for a given monosaccharide. The limit of detection for monosaccharides varied from 0.5 – 10 ug/L, depending on the specific monosaccharide measured. Note that the recovery of monosaccharides from acid hydrolysis of marine samples is typically considered to yield an underestimate of total carbohydrates due to incomplete hydrolysis of acid-resistant structures, and destruction of acid-labile monosaccharides, as well as cross-reactivity of hydrolyzed monosaccharides with amino acids (Engel and Handel, 2011; Becker et al. 2020).

## 140 2.3.4 Polysaccharide extraction for microarray analyses of POM

POM samples were prepared for polysaccharide analysis according to Vidal-Melgosa et al. (2021). Polysaccharides were sequentially extracted from four filter piece punches (11.2 mm diameter) from GF/F filters. The samples were first extracted with autoclaved MilliQ water, followed by 50 mM EDTA, and finally 4 M NaOH with 0.1% NaBH<sub>4</sub>. These different extraction steps are required to extract polysaccharides of varying structure and solubility. The supernatant containing extracted polysaccharides was collected from each of the sequential steps and stored at 4 °C.

#### 2.3.5 Carbohydrate microarray analysis to determine structural complexity of POM

The polysaccharides extracted as described above were analyzed following Vidal-Melgosa et al. (2021). In brief, the polysaccharide extracts were first diluted in printing buffer (55.2% glycerol, 44% water, 0.8% Triton X-100), and then printed on 0.45 µm pore size nitrocellulose membrane (Whatman) using a microarray robot (Sprint, Arrayjet, Roslin, UK) at 20 °C and 50% humidity. The membranes were probed with one of 9 monoclonal antibodies (Table S4), washed multiple times, and probed with secondary antibodies (anti-rat, anti-mouse, or anti-His tag) conjugated to alkaline phosphatase for 2 hours. The arrays were developed using 5-bromo-4-chloro-3-indolyphosphate and nitro blue tetrazolium in alkaline phosphatase buffer (100 mM NaCl, 5 mM MgCl<sub>2</sub>, 100 mM Tris-HCl, pH 9.5). The microarrays were scanned and signal intensity was acquired using the software Array-Pro Analyzer 6.3 (Media Cybernetics). Signals were normalized among samples; higher signals correspond to a higher abundance of a given polysaccharide epitope. Signals were then combined for the different extraction steps, as in Priest et al. (2023), to yield the overall values shown in Fig. 5 and Table S4. Note that the carbohydrate microarray data are only semiquantitative: while comparisons can be made for the abundance of a given epitope among stations and depths, due to fundamental differences in binding intensity of different epitopes to their targets, the signal intensity cannot be used to compare signals of different epitopes.

#### 2.4 Analysis of bacterial community composition

Composition of bacterial communities at all stations and sampling depths was assessed through 16S rRNA gene analysis. Seawater (25 mL) from the Niskin bottles was filtered through 0.22 µm pore size polycarbonate filters at a maximum vacuum of 200 mbar. The filters were air dried and frozen at -20 °C until further processing. Total DNA was extracted from these filters using the DNeasy Power Water Kit (Quiagen). The variable V3 and V4 regions (490 bp) of the 16S rRNA gene were amplified in 30 PCR cycles, using the 5 PRIME HotMasterMix (Quantabio) together with the Bakt\_341F (5' CCTACGGGNGGCWGCAG 3') and Bakt\_805R (5' GACTACGVGGGTATCTAATCC 3') PCR primer pair (Herlemann et al., 2011). Note that these primers omit archaeal sequences; while it is possible that archaea may contribute to enzyme activities, they are excluded from our sequence analyses. Forward and reverse primers were barcoded with individual 8 bp barcode adapter (based on the NEB Multiplex Oligos for Illumina, New England Biolabs). Each amplified PCR product was purified and size selected using the AMPure XP PCR Cleanup system (Beckman Coulter) before barcoded products were

pooled in equimolar concentrations. The pools were sent to the Max Planck Genome Centre (Cologne) for paired-end Illumina sequencing (2x250 bp HiSeq2500). Merging, demultiplexing and quality trimming (sequence length 300–500 bp, 

Stn. 20 was located within the North Atlantic subtropical gyre. Surface water was characteristic of Thermocline Water (Heidrich and Todd, 2020). At a depth of 300 m, T/S characteristics were very similar at Stn. 20 and Stn. 18; this water was likely Eighteen Degree Water (Heidrich and Todd, 2020). Water collected at 3000 m and just above the seafloor had similar T/S characteristics at Stns. 18-20, consistent with North Atlantic Deep Water (Fig. 2; Broecker, 1991).

265

Figure 2: Physical and chemical characteristics of sampling stations within the western North Atlantic. (a) Sea surface temperature, acquired by satellite imaging on/around the sampling day at each station (Rutgers University Coastal Observation Lab: RU-COOL satellite imagery [online] Available from https://marine.rutgers.edu/cool/data/satellites/imagery/(Accessed 28 September 2022). (b) Temperature and salinity diagram of the four stations highlights the differences in water masses sampled. Shapes denote the depths; colors denote the stations.

Leucine incorporation (pM hr-1)

Cell counts (cells mL-1)

Particulate organic carbon (mg C L-1)

Chlorophyll-a (mg m-3)

#### 3.2 Total cell counts and bacterial protein productivity

At all stations, total microbial cell counts were highest in the upper water column and decreased with depth (Fig. 3; Table S2). At Stn. 19, surface and DCM waters had much higher total cell counts than waters at the same depths at Stns. 17, 18, and 20; these high total cell counts coincided with the higher chlorophyll-a concentrations at Stn. 19 (Fig. 3; Fig. S1; Table S2). Cell counts ranged from  $0.3 - 1.0 \times 10^9$  cells L<sup>-1</sup> in surface waters, to ca.  $1.0 - 2.0 \times 10^8$  cells L<sup>-1</sup> at 300 m, to as few as 0.8  $- 1.3 \times 10^7$  cells L<sup>-1</sup> in bottom water at Stns. 18 - 20 (Fig. 3; Table S2).

Bacterial protein productivity—as measured by incorporation of tritiated leucine—was highest in surface waters and decreased with depth. Bulk bacterial productivity (pmol leucine L<sup>-1</sup> hr<sup>-1</sup>) was highest at Stn. 20, where leucine was incorporated at ~2 times the rate as at Stn. 19; this trend holds true for all depths when comparing Stns. 19 and 20 (Fig. 3; Table S2). At Stn. 17, protein productivity increased slightly at the DCM compared to the surface, but was undetectable in bottom waters. At Stns. 18 – 20, bacterial productivity decreased considerably within the first 300 m of the water column, followed by a peak in productivity in lower mesopelagic waters at Stn. 18 and at 1500 m at Stns. 19 and 20, respectively. While bacterial protein productivity roughly doubled at this depth at Stns. 18 and 19, bacterial protein productivity increased even more, 11-fold to 5.5 pmol L<sup>-1</sup> hr<sup>-1</sup> at this depth at Stn. 20. These peaks are followed by decreases in bacterial productivity with depth, and a slight increase again in bottom waters. Bulk bacterial protein productivity in bottom waters increased with distance offshore (Fig. 3; Table S2).

Cell-specific bacterial productivity was calculated by dividing the bulk bacterial productivity for each depth by the cell abundance. Cell-specific bacterial productivity increased in surface waters by two orders of magnitude from the inshore Stn. 17 at  $6 \times 10^{-9}$  pmol cell<sup>-1</sup> hr<sup>-1</sup> to  $5 \times 10^{-7}$  pmol cell<sup>-1</sup> hr<sup>-1</sup> at the offshore, open-ocean Stn. 20 (Fig. S2). Cell-specific bacterial productivity generally increased with distance offshore, with especially high values at Stn. 20. Comparing Stns. 19 and 20, for example, showed that the cell-specific productivity at Stn. 20 was much higher at every depth than at Stn. 19; in surface waters, the bacterial productivity was  $\sim$ 6 times higher at Stn. 20 than at Stn. 19, and  $\sim$ 12 times higher in bottom waters at Stn. 20 than at Stn. 19 (Fig. S2).

#### 3.3 POC concentrations

Overall, POC concentrations were highest in surface and DCM waters, and decreased with depth (Fig. 3; Table S2; Fig. S3). The highest POC concentrations in surface and DCM waters were measured at Stn. 19, the same depths with high chlorophyll fluorescence values (Fig. 3; Table S2; Figs. S1 and S3). In the offshore stations, POC concentrations increased slightly at bottom depths compared to the depths measured above.

## 3.4 Monosaccharide composition of POM-derived combined carbohydrates

The monosaccharide constituents of POM-derived combined carbohydrates were very similar in surface and DCM waters of all stations (ANOVA, p = 0.90), but changed markedly with depth (Fig. 4). In subsurface waters, these constituents were significantly different from the constituents in surface and DCM waters (ANOVA, p = 0.026). In surface waters at all stations, glucose, xylose, and galactose composed ~80% or more of the total combined carbohydrates. Although fucose, galactosamine, arabinose, glucosamine, and glucuronic acid composed ~20% or less of total combined carbohydrates, they were more abundant in surface waters and decreased considerably in relative abundance with depth (Fig. 4). Galactosamine and arabinose were only detected in surface and DCM waters offshore, and at all depths at Stn. 17. Overall, the concentration of POM-derived combined carbohydrates in seawater decreased with depth as well, ranging from 6-28  $\mu$ g/L in surface waters to ~0.5-1  $\mu$ g/L in bottom waters (Fig. 4).

In the upper water column, total concentrations of monosaccharides contributing to combined carbohydrates differed somewhat by station (Fig. 4b; ANOVA, p = 0.014). Stn. 19 had the highest concentration (~28 µg L<sup>-1</sup> in surface waters), consistent with the high chlorophyll fluorescence (Fig. S1) and high POC concentration at this station (Fig. 3). At Stns. 18 and 20, the concentrations were much lower (~6 µg L<sup>-1</sup>). Stn. 17 (with a total water column depth of ca. 600 m) also had higher total concentrations in both surface and bottom waters, as well as different monosaccharide constituents than at the other stations (Fig. 4). At Stns. 18-20, there were no station-specific differences in monosaccharide composition in waters below DCM (ANOVA, p = 0.58).

Figure 4: Monosaccharide constituents of the combined carbohydrates in POM. (a) The relative contribution of the combined carbohydrates shows the percentage contribution of each monosaccharide to the total. (b) The concentration of each monosaccharide, as determined by HPAEC-PAD.

## 3.5 Polysaccharide structures of POM detected via microarray analysis

The carbohydrate epitopes—and therefore the structural complexity of polysaccharides in POM—differed among stations (Fig. 5; ANOVA, p = 0.016). Although filtered samples from all depths and stations were included in the epitope analyses (Table S3; Table S4), polysaccharide structures extracted from POM were only detected in surface and DCM waters, with the exception of the  $\beta$ -1,4-mannan detected in the comparatively shallow bottom water at Stn. 17 (Fig. 5; Table S4). At all stations,  $\beta$ -1,3-glucan and alginate were detected in surface and DCM waters. These were the only epitopes detected at Stn. 18; at all other stations, additional epitopes were detected. At Stn. 17, fucoidan (epitope II) was also detected in surface waters, and  $\beta$ -1,4-mannan in bottom waters at 625 m. In the highly productive surface waters of Stn. 19 (Fig. 3, Table S2), all of the epitopes except fucoidan (epitope II) were detected (Fig. 5). At Stn. 20, in addition to  $\beta$ -1,3-glucan and alginate, fucoidan (epitopes I and II) were detected, as was  $\beta$ -1,4-mannan.

Figure 5: Heatmap of polysaccharides detected from the carbohydrate microarrays. Darker shades represent a higher relative intensity, while lighter shades represent a lower relative intensity; white spaces represent samples where no epitopes were detected. Note that samples were collected at each depth horizon for all stations, but depths below the DCM and above bottom waters are not explicitly shown on the Y axis in the figure, as no epitopes were detected in any of those samples.

#### 3.6 Community composition

Sequencing of 16s rRNA genes was intended to provide an overview of bacterial community composition. Since particle-associated bacteria are typically a small fraction of the total community (Bachman et al., 2018), our sequences – although carried out with water that was not prefiltered – likely primarily reflect the free-living bacterial community. It should also be noted that different primers can yield different results, especially at the ASV level, and results of community

composition based on 16s rRNA gene sequencing frequently differ in part from direct counts of cells carried out by FISH (e.g., Fadeev et al., 2021). Even considering these caveats, however, bacterial community composition differed considerably by depth (ANOSIM, R-stat = 0.3248, p = 0.0052; Fig. 6), and also to an extent by station (Fig. S4). In surface and DCM waters at all four stations, bacterial communities were mainly composed of members of the Proteobacteria (Gamma- and Alphaproteobacteria), Bacteroidota, Actinobacteria, and Cyanobacteria. In waters at 300 m and below, higher relative abundances of the SAR324 clade, Marinimicrobia, Chloroflexi and Firmicutes were detected; Bacteroidota and Actinobacteria were present in lower relative numbers, and Cyanobacteria had virtually disappeared (Fig. S5).

Stns. 17 and 18 surface and DCM waters were distinguished by a large contribution (~30%) of Prochlorococcus, which was not observed at Stns. 19 and 20 (Fig. S5). Surface waters of Stn. 19 were dominated by Alphaproteobacteria; members of the Bacteroidota were present in greater abundance with a wider array of taxa than at other locations. In Stn. 19 DCM waters, Gammaproteobacteria made up a larger portion of the bacterial community (~30%) compared to other stations. Stn. 20 surface waters were dominated by *SAR11 clade Ia* (Fig. S5).

Figure 6: Non-metric multidimensional scaling (NMDS) plots based on Bray-Curtis dissimilarity shows bacterial communities clustered by depth. Stations are represented by shapes; depths are indicated by colors (see key). The grey shaded regions in the NMDS plots represent the clustering of bacterial communities by depth.

Bacterial communities in subsurface waters differed substantially from their surface counterparts (ANOSIM, R-stat = 0.3248, p = 0.0052; Fig. 6; Fig. S5). Stn. 17 bottom water (at 625 m) had a strong representation of Gammaproteobacteria (~35% of the community), while the contribution of Prochlorococcus and Alphaproteobacteria decreased relative to the surface and DCM. At Stn. 18, the communities at 300 m, the lower mesopelagic, and 1500 m were relatively similar to each other,

and also to communities in the lower mesopelagic and 1500 m at Stn. 19. These communities were dominated by Alphaproteobacteria and SAR324. Stn. 20 communities at the lower mesopelagic and in deeper waters were distinct from communities at similar depths for Stns. 18 and 19.

Bacterial communities at 3000m and in bottom water differed considerably among stations (Fig. 6; Fig. S5). The bottom water community at 3190 m at Stn. 18 stands out based on the contribution of Bacteroidota, as well as the presence of *Prochlorococcus*. Bacteroidota were also detected in 3000 m and bottom waters at Stn. 19, although this phylum was in low relative abundance at depths between 300 to 1500 m. Stn. 19 bottom waters were distinguished by the contributions of SAR324, Marinimicrobia, Chloroflexi, and Firmicutes, which constituted ~50% of the bacterial community at this depth. In Stn. 20 bottom waters, the Gammaproteobacteria—and particularly *Pseudoalteromonas*—accounted for up to 40 % of the community.

# 3.7 Glucosidase and peptidase activities

Peptidase and glucosidase activities differed by station and depth (Fig. 7; Fig. S6). Leucine aminopeptidase activity (exo-peptidase, a terminal unit cleaving enzyme) tended to dominate at all depths at Stn. 19, whereas at the other stations, endopeptidase activities (activities of mid-chain cleaving enzymes) were frequently comparable to leucine aminopeptidase activities. Alpha- and beta-glucosidase activities were particularly prominent at 300 m and in the lower mesopelagic (Fig. 7).

In general, summed glucosidase and peptidase activities at 3000+ m depths were only slightly lower than at shallower depths. Summed activities were highest at surface and DCM waters (upwards of 150-200 nmol L<sup>-1</sup> hr<sup>-1</sup>), then decreased by half (to upwards of 50-100 nmol L<sup>-1</sup> hr<sup>-1</sup>) below 300 m (Fig. S7). Fewer substrates were typically hydrolyzed at deeper depths, although Stn. 20 showed hydrolysis of all seven glucosidase and peptidase substrates at 1500 m as well as in bottom water (Fig. 7). At Stn. 18, one endopeptidase, chymotrypsin substrate AAPF, was hydrolyzed more rapidly at deeper depths (depths at the lower mesopelagic and below), with particularly high activity in bottom water. In contrast, leucine aminopeptidase activities were generally high at all depths at Stn. 19, and was the highest peptidase activity at depths of 1500m and below at Stn. 20.

Figure 7: (a) Average glucosidase and peptidase activities for each station and depth; error bars represent the standard deviation of activities measured over 12 hours. Note that y-axes differ between each depth. A-glu =  $\alpha$ -glucosidase; b-glu =  $\beta$ -glucosidase; leu = Leucine aminopeptidase; AAF = alanine-alanine-phenylalanine; AAPF = alanine-alanine-phenylalanine; QAR = glutamine-alanine-arginine; FSR = phenylalanine-serine-arginine. Non-metric multidimensional scaling (NMDS) plot based on Bray-Curtis dissimilarity shows (b) peptidase and (c) glucosidase activities clustered by depth. Shapes represent stations; colors represent depths. The shaded regions represent the grouping of enzyme activities by depth.

#### 3.8 Polysaccharide hydrolase activities

Polysaccharide hydrolase activities varied considerably by depth (Fig. 8); the NMDS plot showed broad overlaps among stations (Fig. S8). The spectrum of enzyme activities (the number of different polysaccharides hydrolyzed) was generally broadest not in surface waters, where three or fewer polysaccharide substrates were hydrolyzed, but at the DCM, at 300 m, in the lower mesopelagic (ca. 800 m), or at 1500 m, where generally four or five substrates were hydrolyzed (Fig. 8). Stn. 18 lower mesopelagic and Stns. 19 and 20 1500 m waters showed increased polysaccharide hydrolase activities compared with other stations at the comparable depths; these higher activities are also reflected in the bacterial productivity data (Fig. 3). Summed polysaccharide hydrolase activities were lower in 3000+ m waters (~1-4 nmol monomer L-1 hr-1) compared to surface, DCM, 300 m, and lower mesopelagic waters (~4-40 nmol monomer L-1 hr-1; Fig. S9). At deeper depths, enzymatic activities were also generally first detected at later timepoints (Fig. 8).

Figure 8: (a) Polysaccharide hydrolase activities for each station and depth; the hydrolysis rates for each timepoint show when each substrate was hydrolyzed. Note that y-axes differ between each depth. Pul = Pullulan; Lam = Laminarin; Xyl = Xylan; Fuc = Fucoidan; Ara = Arabinogalactan; Chn = Chondroitin Sulfate. (b) Non-metric multidimensional scaling (NMDS) plot based on Bray-Curtis dissimilarity shows polysaccharide hydrolysis rates clustered by depth. Shapes represent stations; colors represent depths. The shaded regions represent the grouping of polysaccharide hydrolase activities by depth.

Hydrolysis of pullulan, laminarin, chondroitin, and xylan was measurable at most stations and depths. Laminarin hydrolysis was measured in 19 of 23 distinct depths across stations, pullulan hydrolysis was measured in 18 depths across stations, xylanase activity was measured in 16 depths across stations, and chondroitin sulfate was hydrolyzed at 15 depths across stations. Fucoidan and arabinogalactan hydrolysis was rarely detected; fucoidan was only measurably hydrolyzed at 300 m at Stns. 18 and 20, and arabinogalactan was only measurably hydrolyzed in Stn. 19 surface waters and, notably, at Stn. 20 at 1500 m. No stations or depths showed hydrolysis of all six polysaccharides.

#### 3.9 Correlations among environmental parameters, carbohydrates, and enzymatic activities

Connections among enzymatic activities, environmental parameters, and carbohydrate composition across all stations were investigated using a corrplot based on Pearson correlations (Fig. 9). Note that specific bacterial taxa were not included in this corrplot because the polysaccharide hydrolase complement of bacteria varies widely even among closely related taxa (e.g. Avci et al., 2020, Krüger et al., 2019); moreover, resource prioritization can mean that even bacteria with the capability of hydrolyzing specific polysaccharides may target only a subset of the polysaccharides that they are capable of using (Koch et al., 2019). Total cell counts (TCC) correlated strongly with particulate organic carbon, chlorophyll-*a*, and carbohydrate content (i.e., the constituent monosaccharides of POC), including all monosaccharides except muramic and glucuronic acids. Most monosaccharides—again, with the exception of muramic acid and glucuronic acid—were also highly correlated with one another (Fig. 9), as were those same monosaccharides and the polysaccharide epitopes to detect β-1,3-glucan, β-1,4-mannan, xylosyl residues, and alginate. Correlations among enzyme activities were evident in only a few cases: laminarinase with pullulanase; xylanase and fucoidanase. QAR and FSR were highly correlated with one another; β-glucosidase was positively correlated with xylanase.

Figure 9: Correlation plot displaying Pearson's correlations among enzymatic activities, environmental parameters, and carbohydrate constituents for all samples. Blue denotes positive correlations while red denotes negative correlations. The shade and size of the circle represents the intensity of correlation. \* denotes statistical significance (p 

Figure 10: Correlation plots displaying Pearson's correlations between enzymatic activities in different depth zones. Epipelagic zone: 460 surface and DCM waters; Mesopelagic zone: 300 m, 800/850 m, and 1500 m; Bathypelagic zone: >1500 m. Blue denotes positive correlations while red denotes negative correlations. The shade and size of the circle emphasizes the intensity of correlation between enzymatic activities. (?) for a given column or row represents a lack of data to assess a correlation for a given observation or measurement. \* denotes statistical significance (p < 0.05) after Bonferroni correction.

#### 4 Discussion

The production and decomposition of marine organic matter is a function of interlinked oceanic processes and actors: marine productivity is powered by phytoplankton at the base of the food web that produce high molecular weight biopolymers such as polysaccharides and proteins. These biopolymers can take both dissolved (DOM) and particulate (POM) forms, and include DOM excreted by phytoplankton and algae during growth (e.g., Mitulla et al., 2016; Buck-Wiese et al., 2023), as well as DOM released via viral lysis and grazing (Suttle, 2005; Middelboe, 2008; Nagata and Kirchman, 1992). POM by definition includes phytoplankton and other organisms that are retained via filtration, as well as POM produced via aggregation and gel formation of high molecular weight DOM - especially polysaccharide-containing DOM (Zhou et al., 1998; Verdugo et al., 2004; Huang et al., 2021) - that can entrain smaller particles into larger sinking aggregates (Markussen et al., 2020; Vidal-Melgosa et al., 2021). Marine organic matter therefore is present in a continuum of sizes and forms (Azam and Malfatti, 2007; Iversen 2023). All of this organic matter fuels heterotrophs, including the heterotrophic bacteria that cycle a large fraction of marine productivity (Azam and Malfatti, 2007); this organic matter must also be hydrolyzed to sufficiently small sizes prior to bacterial uptake. The nature and structure of this organic matter - whether dissolved or particulate - therefore dictate the enzymatic tools needed by heterotrophic microbial communities to access them (Gügi et al., 2015; Thomas et al., 2017). Microbial communities vary in their summed genetic capabilities and the extent to which they can activate them at a specific time and place; thus, substrates that are labile in one location may be recalcitrant in another (Arnosti et al., 2021). Untangling these interrelationships—the structure of the biopolymers found in different fractions of organic matter, the composition and capabilities of the heterotrophic communities, and the activities of the specific enzymes they produce—is essential to understand the rate and location at which organic matter is cycled in the ocean (Fig. 1a).

#### 4.1 Carbohydrates, microbial communities, and their enzymatic capabilities by site and depth

The constituent monosaccharides of POM were similar across stations, showing slight differences between the upper water column (surface and DCM) compared to deeper depths (Fig. 3), a pattern previously observed in other locations in the ocean (Handa and Tominaga, 1969; Handa and Yanagi, 1969). This similarity in constituent monosaccharides (Fig. 1b), however, masks considerable differences in the carbohydrate epitopes detected in the POM collected in surface and DCM waters (Fig. 5). The number of epitopes detected ranged from a high of seven at Stn. 19 to a low of two at Stn. 18 (Fig. 5, Table S4); at all locations,  $\beta$ -1,3-glucan and alginate were detected. The widespread detection of  $\beta$ -1,3-glucan in our samples could be linked to the high prevalence of laminarin in the ocean (Alderkamp et al. 2007; Becker et al. 2020); this epitope has

also been detected in cultured diatoms and diatom-derived HMW DOM (Huang et al., 2021), in POM and HMW DOM during a spring sampling series in the North Sea (Vidal-Melgosa et al., 2021), and in POM from all surface water samples at a series of sites around Svalbard (Priest et al., 2023). Alginate was also widely detected via epitopes in POM from surface water samples from Svalbard (Priest et al., 2023), in exudates of macroalgae (Koch et al., 2019), and has additionally been detected in cores collected from anoxic brine sediments in the Red Sea (Vidal-Melgosa et al., 2022). Although the alginate probe (BAM-7; Table S4) has been reported to cross-react with fucose-containing polysaccharides (Torode et al., 2016), the observation that alginate was detected in our samples at depths and locations where no fucoidan was detected (e.g., Stn. 18; DCM of Stns. 17 and 20) suggests that in these cases, at minimum, the positive signal was not due to cross-reaction. Fucoidan was detected, however, in surface waters of Stns. 17 and 20 (Epitope II; plus Epitope I for Stn. 20), as well as surface and DCM waters of Stn. 19 (Epitope I). These epitopes could be reacting to diatom-derived fucoidan (Huang et al., 2021; Vidal-Melgosa et al., 2021). Stn. 19 waters were notable in that most of the epitopes investigated were detected at this station: in addition to β-1,3glucan and alginate, rhamnogalacturonan, \(\text{B-1.4-mannan}\), and xylosyl residues were detected in POM from surface and DCM waters, and β-1,4-glucan in POM from surface waters (Fig. 5; Table S4). This wide array of epitopes suggest that a considerable diversity of complex polysaccharides was present in POM samples at Stn. 19, a diversity that may be due to the North Atlantic spring bloom (high chlorophyll-q concentrations; Fig. 3), with a high concentration of phytoplankton. In support of this hypothesis, sequential waves of phytoplankton and a considerable array of carbohydrate epitopes have been reported in POM as well as HMW DOM during spring blooms in the German Bight (Vidal-Melgosa et al. 2021). In contrast, at Stn. 18, in the Gulf Stream, just two epitopes were detected in surface and DCM waters (Fig. 5). The differences in polysaccharide complexity detected in POM from Stns. 18 and 19 could be driven in part also by the different phytoplankton communities that dominate distinct regions of the western North Atlantic (Bolaños et al., 2020; Della Penna and Gaube, 2019).

In accordance with the structural differences of POM indicated by carbohydrate epitopes, activities of extracellular enzymes—which are also very sensitive to substrate structural features—differed notably by station and depth (Fig. 8; ANOVA, p = 0.0011). To measure potential activities of polysaccharides and proteins, two key components of POM (Hedges et al., 2002) and HMW DOM (McCarthy et al., 1996), we measured hydrolysis of a series of substrates that vary in structural complexity as well as in enzyme class. We note that the enzyme activities were measured in comparatively small volumes of water, and thus likely primarily represented enzymes of free-living microbial communities; since bacterial community composition was measured in water volumes comparable to those used for the enzyme activity measurements, free-living microbial communities likely also dominated the sequences we obtained. However, a previous investigation in other parts of the western North Atlantic Ocean demonstrated that enzyme activities measurable in volumes of unfiltered water comparable to those used in the current study were always also measurable in the particle-associated fraction (Lloyd et al., 2022). In any case, polysaccharide hydrolase activities show spatial variations that parallel spatial differences in microbial community composition (Figs. 6; 8b; Arnosti et al., 2011; 2012); depth-related patterns have also been observed (Steen et al., 2012; Hoarfrost and Arnosti 2017; Balmonte et al., 2021; Lloyd et al., 2023). We note, however, that drawing a more direct

connection between specific organisms and the activities of specific enzymes - with the exception of selfish bacteria (Cuskin et al., 2015; Reintjes et al., 2017) – is currently beyond the state of the art.

We also examined the statistical connection among polysaccharide- and protein-hydrolyzing enzymes, as well as physical and environmental factors. Among the four stations, a handful of very robust correlations in enzyme activities were evident (Fig. 9), but there were also considerable differences among individual stations (Fig. S9). These variations in activities likely reflect the considerable variety of enzymes produced even by closely related bacteria (Avci et al., 2020; Krüger et al., 2019), as well as resource prioritization for specific substrates among bacteria (Koch et al., 2019). Given the depth stratification of microbial communities (Fig. 6; Fig. S5), which extends to the ability of communities to excrete extracellular enzymes (Zhao et al., 2020), we investigated correlations among enzyme activities by depth zone (Fig. 10). This corrplot revealed patterns not evident from the depth-integrated corrplots (Fig. 9). Most notably, in the mesopelagic, most of the polysaccharide hydrolases positively correlate with one another, despite the fact that different enzymes, often produced by different organisms, are responsible for hydrolysis of these structurally-distinct polysaccharides (e.g., Becker et al., 2020; Reisky et al., 2019; Sichert et al., 2020). Peptidase correlations, however, were not notably more prevalent in the mesopelagic than in other depth zones. These correlations suggest that in the mesopelagic, polysaccharides—including those with more complex structures—are a specific target, in accordance with studies that have demonstrated intense organic matter remineralization in this zone (Boyd et al., 1999; Buesseler et al., 2007).

A surprising observation is that activities of the exo-acting enzymes ( $\beta$ - and  $\alpha$ -glucosidase, leucine aminopeptidase) seldom showed positive correlations with endo-acting enzymes targeting the same substrate class (i.e., leucine aminopeptidase with the other peptidases: AAF, AAPF, QAR, and FSR;  $\beta$ - and  $\alpha$ -glucosidase with the polysaccharide hydrolases). Moreover,  $\alpha$ - and  $\beta$ -glucosidase activities did not show positive correlations with each other. Leucine aminopeptidase activity overall strongly correlated with most of the monomers of combined carbohydrates, as well as some of the carbohydrate epitopes (Fig. 9), perhaps a sign of activity hydrolyzing terminal amino acids from polysaccharide components, given that leucine aminopeptidase activities the activities of multiple exo-acting peptidases (Steen et al., 2015). However, leucine-aminopeptidase activities did not otherwise show broad-scale correlations with the endopeptidase activities. The lack of correlations between activities of the exo- and endo-acting enzymes targeting carbohydrates, as well as those targeting peptides, suggest that these enzymes may be targeting structures that include terminal monosaccharides or amino acids. In any case, these results suggest that measurements of exo-acting activities with leucine-MCA and MUF- $\beta$ -glucosidase should not be used generally as overall proxies for polysaccharide or protein degradation by microbial communities.

Spatial differences in polysaccharide hydrolase activities (Fig. 8) are likely a function of depth- and location-related differences in microbial community composition (Fig. 6; Fig. S5). These differences have been linked to substrate availability, as in the case of the shifting nature of organic matter produced during phytoplankton blooms (e.g. Teeling et al., 2012; Dlugosch et al., 2023), with specific organisms targeting distinct polysaccharide structures (Francis et al., 2021; Giljan et al., 2023; Orellana et al., 2022). While depth-stratification in microbial composition has been previously reported (e.g., DeLong

et al., 2006), including in the northwestern Atlantic Ocean (Zorz et al., 2019), we found indications of differences in community composition across locations (Fig S4; Fig. S5). Location-related differences were particularly pronounced in the surface ocean (Fig. S4), where physical, biological, and chemical properties of the water masses differed the most (Figs. 2, 3; Table S2). For example, the high chlorophyll fluorescence at Stn. 19 at the time of sampling likely is due to the North Atlantic Spring Bloom; phytoplankton blooms have been shown to lead to increases in bacteria that possess the genes capable of degrading complex substrates (e.g. Teeling et al. 2016; Kalenborn et al., 2024). Differences in community composition and enzymatic function (Figs. 6, 8) were also evident in the water below 1500 m from all stations, despite all water from these depths having characteristics typical of North Atlantic Deep Water (Broecker, 1991; Fig. 2). These differences in community composition and enzymatic function could be due in part to the ca. 2000 m difference in bottom water depths between Stns. 18 and 20 (Table S2), as well as from differences in the quantity and nature of sinking particles at each location (Mestre et al., 2018; Pelve et al., 2017), which may also affect enzymatic activities in the deep ocean (Lloyd et al., 2022).

# 4.2 Polysaccharide structural complexity could in part explain patterns of heterotrophic carbon cycling in the ocean

This analysis of carbohydrate epitopes in POM, only the second study using this technique on open ocean samples (Priest et al., 2023), provides insight into the composition of intact polysaccharides in the upper ocean. The presence of 1,3-β-D-glycans and alginate at all stations in surface and DCM waters (Fig. 5) demonstrates that laminarin and alginate-derived structures are found in POM from a broad range of locations. At Stns. 17, 19, and 20, moreover, at least one of the fucoidan epitopes was detected in POM from surface waters. Detection of these structures is particularly intriguing, since laminarin and fucoidan in essence are at the opposite ends of the degradation spectrum. Laminarin is rapidly hydrolyzed in most ocean waters, but fucoidan hydrolysis is rarely detected in the water column (Arnosti et al., 2011; Hoarfrost and Arnosti, 2017; Balmonte et al., 2021; Lloyd et al., 2023). This pattern holds for the current investigation as well, where fucoidan hydrolysis was only measurable at 300 m at Stns. 18 and 20, whereas laminarin hydrolysis was widely detected, at 19 of the 23 stations and depths (Fig. 8).

The difference in hydrolysis of laminarin and fucoidan is likely linked to substrate structural complexity, since the number of different types of enzymes required to hydrolyze a polysaccharide scales linearly with polysaccharide structural complexity (Bligh et al., 2022); fucoidan in particular represents a hydrolytic challenge (Sichert et al., 2020). POM in the upper ocean likely includes polysaccharide structures that are highly labile, as well as those that are generally recalcitrant with respect to bacterial remineralization. We surmise that with increasing depth in the ocean, the contribution of labile constituents would decrease, and the relative contribution of recalcitrant polysaccharides would be enhanced. Testing this hypothesis would require investigation of polysaccharide epitopes in POM collected in deep water, where sinking particles can be substantially re-worked by bacterial communities. In the current investigation, however, aside from the (shallow) bottom water at Stn. 17, we were not able to detect polysaccharide epitopes in POM from the deep ocean. Given the low concentrations of POM in the deep ocean (Table S2; Baker et al., 2017), the potential for incomplete extraction of polysaccharides by our methods, and the comparatively small volume of water we filtered for POM analysis, we surmise that any polysaccharides present in deeper

samples were below our limit of detection. Alternatively, deep ocean POM may not be characterizable using epitopes due to bacterial transformations of sinking particles (Wakeham et al., 1997; Hedges et al., 2001; Kharbusch et al., 2020). However, previous studies have also reported rapid fluxes of fresh organic matter to the deep ocean (Follett et al., 2014; Ruiz-Gonzalez et al., 2020; Poff et al., 2021), suggesting that labile polysaccharides may be present even in deep waters. In support of this point, bathypelagic bacteria capable of selfish uptake of a broad range of polysaccharides (including fucoidan) have recently been identified in deep ocean waters at these same stations (Giljan et al., 2023). Because this mechanism of acquiring polysaccharides involves initially binding, partially hydrolyzing, and transporting larger macromolecules to the periplasmic region of the cell (Cuskin et al., 2015), this finding suggests that intact polysaccharides are components of fresh organic matter that reaches the bottom of the ocean (Giljan et al., 2023). Collection of larger quantities of POM, and/or improvements in extraction methods, may lead in the future to characterization of intact polysaccharides in deep ocean POM.

The structural complexity of polysaccharides may help explain patterns of the enzymatic capabilities of microbial communities and the *in-situ* carbohydrate signatures that differ among the stations, information that is not evident when analyzing the individual monosaccharide building blocks of combined carbohydrates (Fig. 1b). This variability may not exist to the same extent in the peptidase activities (Fig. 7), since peptidases generally have a broader array of target substrates (Lapébie et al., 2019). However, the observation that activities of both classes of exo-acting enzymes (glucosidases and leucine aminopeptidase) do not correlate with the activities of endo-acting enzymes (polysaccharide hydrolases and trypsin- and chymotrypsin peptidases; Figs. 9, 10) that target the same class of molecule suggests that hydrolysis of HMW substrates involves a level of structural complexity that is not easily captured experimentally (Fig 1). Epitope analysis begins to reveal some of these otherwise hidden structural features of polysaccharides, providing a pathway to understand the intricacies of the microbial community's enzymatic toolbox and the manner in which microbial activities vary with location and depth in the ocean.

# **Competing Interests**

The contact author has declared that none of the authors has any competing interests.

## Acknowledgments

We thank the captain and crew of the R/V *Endeavor*, as well as the members of the scientific party of EN638, for excellent work at sea. We would like to thank John Bane for valuable discussions pre- and post-cruise, and for his help interpreting the physical oceanography of the region. Funding was provided by the U.S. National Science Foundation (OCE-2022952 to CA), with additional funding from the Max Planck Society.

## **Author Contributions**

CCL: Investigation, Formal analysis, Data curation, Visualization, Writing – Original Draft, Writing – Review and Editing;
 SB: Investigation, Data curation, Methodology, Writing – Original Draft; Writing – Review and Editing;
 GG: Investigation, Data curation, Visualization, Writing – Original Draft; Writing – Review and Editing;
 SV-M:Investigation, Data curation, Methodology, Writing – Review and Editing;
 NS: Investigation, Writing – Review and Editing;
 GG: Investigation, Data curation, Methodology, Writing – Review and Editing, Funding acquisition;
 GG: Investigation, Methodology, Writing – Review and Editing, Funding acquisition;
 Ga: Investigation, Methodology, Investigation, Writing – Review and Editing, Supervision, Funding acquisition;
 CA: Conceptualization, Methodology, Investigation, Writing – Original Draft, Writing – Review and Editing, Supervision, Project

## Data availability

administration, Funding acquisition

- The polysaccharide hydrolase dataset can be accessed via the Biological and Chemical Oceanography Data Management Office (BCO-DMO) under the Project: A mechanistic microbial underpinning for the size-reactivity continuum of dissolved organic carbon degradation (Microbial DOC Degradation; doi: 10.26008/1912/bco-dmo.821801.1).
  - The peptidase and glucosidase extracellular hydrolase activity dataset can be accessed via the Biological and Chemical Oceanography Data Management Office (BCO-DMO) under the Project: A mechanistic microbial underpinning for the size-reactivity continuum of dissolved organic carbon degradation (Microbial DOC Degradation; doi: 10.26008/1912/bco-dmo.820973.1).
- The total cell abundance dataset can be accessed via the Biological and Chemical Oceanography Data Management Office (BCO-DMO) under the Project: A mechanistic microbial underpinning for the size-reactivity continuum of dissolved organic
- The bacterial productivity dataset can be accessed via the Biological and Chemical Oceanography Data Management Office (BCO-DMO) under the Project: A mechanistic microbial underpinning for the size-reactivity continuum of dissolved organic carbon degradation (Microbial DOC Degradation; doi: 10.26008/1912/bco-dmo.820556.1).

carbon degradation (Microbial DOC Degradation; doi: 10.26008/1912/bco-dmo.820961.1).

- The monosaccharide constituents of particulate organic matter dataset can be accessed via Zenodo using the following doi: 10.5281/zenodo.15375159.
- The carbohydrate epitopes of particulate organic matter dataset can be accessed via Zenodo using the following doi: 10.5281/zenodo.15375738.
- Bacterial community composition can be accessed through the European Nucleotide Archive (ENA) under accession number PRJEB63119.

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
