# Peer review of "Carbohydrates, enzyme activities, and microbial communities across depth gradients in the western North Atlantic Ocean"

_EGUsphere, 2025_

## Author Response (AR1)

RC1: 'Comment on egusphere-2025-2249', Anonymous Referee #1, 07 Jul 2025 reply

**Remarks to the Author:**

This manuscript presents important findings about the composition and distribution of polysaccharides in particulate organic matter (POM) and discusses the relationship between polysaccharide structure, hydrolytic enzyme activity, and bacterial community composition through the water column across the western North Atlantic Ocean. The diverse combination of monosaccharides and linkages in polysaccharides necessitates a wide array of substrate-specific enzymes to hydrolyze them. Thus, the lability of polysaccharides in the environment is related to the presence of hydrolytic enzymes and the bacterial community that produces them.

In this study, the authors characterize the polysaccharide composition in POM from up to 7 depth horizons at four stations in the western North Atlantic. They measured monosaccharide composition, which was similar across all stations, but varied with depth. Carbohydrate epitopes provided evidence of polysaccharide diversity between stations with greater carbohydrate diversity being measured at the station with the highest chlorophyll-a concentration. This study also examined the distribution of hydrolytic enzyme activity in the seawater and the bacterial community composition. Specific enzyme activity differed between stations and with depth. Some enzymes, such as pullulan and laminarin hydrolysis enzymes, were active in most samples while other enzymes, such as arabinogalactan and fucoidan hydrolysis enzymes, were only active at a few depths and at a few stations. This patchiness was also seen in the bacterial community composition, a nice result which demonstrates that community composition impacts the distribution of some enzymes in the ocean.

The study follows a logical path and leads the reader to clearly stated conclusions. The array of methodologies employed in this study is impressive and a considerable thought has clearly gone into the data analysis. However, I have several major concerns regarding the comparative analyses and conclusions of the paper that need to be addressed before publication. I think this work is a good fit for Biogeosciences, and should be strongly considered for publication once the concerns below are addressed.

We thank the reviewer for the very thoughtful and thorough comments that greatly improved our manuscript.

**General Concerns:**

1) The title of the paper, the abstract, and the heading of section 4.1 are misleading, as they suggest that the paper will link the distribution of different carbohydrates to the distribution of enzyme activities and the bacterial community composition. In fact, in section 4.1 each of these datasets is discussed independently.

We retitled the manuscript to "Carbohydrates, enzyme activities, and bacterial communities across depth gradients in the western North Atlantic Ocean" so that the major data elements are clear from the title, but do not create the impression that we will be presenting explicit links that

at this point are beyond the current state of the art. We also re-wrote the abstract and changed the heading of section 4.1.

The authors only state that the differences in polysaccharide complexity could be due to the differences in the microbial community.

I recognize that no current method exists for linking hydrolytic enzyme activity with the presence of specific taxa, as the authors rightfully point out in the introduction. But the framing of the paper leads the reader to believe that some relationship between polysaccharide composition and enzymatic activity is going to be developed. This is made more difficult because the enzymatic hydrolysis data and the carbohydrate epitope data come from two fundamentally different datasets: the carbohydrate composition and epitope data, which come only from POM, and the bacterial community composition and hydrolytic enzyme activity data, which come from unfiltered seawater. All of this leaves the reader unsure of the key conclusions of the paper. I suggest the authors better define the sections in the discussion to highlight the key conclusions of the paper. These can then be used to adjust the abstract accordingly.

In the Introduction, we briefly discuss the differences between POM and HMW DOM. In the Discussion, we went into these differences in a little more depth, and completely rewrote the next sections of the Discussion, with a major focus on the carbohydrate epitopes and on carbohydrates in POM. We discussed the point that our bacterial community composition data and enzyme activities likely primarily reflect the activities of free-living bacteria; we also note that a previous study in this same area found that all of the activities that were measurable in the free-living fraction (similar sample volumes as the present study) were also measurable in the particle-associated fraction.

2) Throughout the manuscript, the authors discuss HWMDOM or "organic matter," even though their carbohydrate data is for POM only (ln 13, 34, 69, Fig.1). This has the effect of confusing the reader, as the link between POM and HMWDOM is not fully known.

In the Introduction as well as in the Discussion we better outlined which fractions were measured.

I recognize that the carbohydrate epitope analysis was only able to be conducted on POM due to the need for sample mass, and that the authors would like to expand their analyses and conclusions to organic matter more broadly. But the way in which the authors switch between discussing POM, HMWDOM, and "organic matter" leads to confusion. I recommend that the authors confine their study to discussing POM, for which there is a vast existing body of literature, to which this work would contribute important new information.

It would be appropriate for the authors to compare their carbohydrate epitope and monosaccharide composition data to HMWDOM in a clearly defined and concise section of the discussion. Doing so would help define the bounds of this study, and the comparison would enhance this work while also broadening its readership.

We focused much of the Discussion on the carbohydrate composition of POM, and included a considerable number of new references to make our points.

3) The authors use several NMDS plots to visualize differences between among stations and depth horizons for several of the datasets presented. Many of the NMDS plots, especially the ones that show clustering by station, do not appear to show any differences between the clusters. The authors should consider if each NMDS plot adds to the paper in its current form.

As an example in the text of section 3.6, the authors note that Prochlorococcus was in high abundance in the surface and DCM waters of station 17 and 18, but not in 19 or 20. This trend appears on the NMDS plots with the surface and DCM points of stations 17 and 18 clustering near one another, but the surface and DCM points of stations 19 and 20 not clustering. This observation is obscured, however, by the grey shaded areas, which do not show this detail. The authors should consider if drawing clusters by depth and station is the best way of showing compositional similarities or differences.

We moved the station-related NMDS plots to the SI, and include a plot of the depth-related NMDS that does not show the gray hulls.

**Specific Comments:**

Ln 21: You do not demonstrate that the structure of the microbial community is interrelated with the complexity of carbohydrates in POM. You show that both vary with depth and location, but not that those variations are significantly correlated.

We rewrote this sentence.

Ln 65: Vidal-Melgosa et al., (2021) use this technique on HMWDOM, so why do you say it can only be carried out on POM? Please clarify this statement.

We miswrote this sentence: epitope detection as currently carried out does not work on DOM if it is not concentrated to yield the HMW-DOM fraction, as it was in Vidal-Melgosa et al. 2021. This point is now stated in the MS.

Ln 93, 465, and elsewhere: As someone who does not work on polysaccharide-degrading enzymes, it would be useful to know what the distinction between a polysaccharide hydrolase and a glucosidase is. A short sentence either in the introduction or in the results would help.

We added a new section to clearly distinguish among enzyme types and activities.

Ln 131: Do you have a metric for monosaccharide recoveries from acid hydrolysis?

One of the issues (generally) in acid hydrolysis of samples for monosaccharide analyses is that the conditions required to free some monosaccharides from polysaccharides can destroy other monosaccharides. Becker et al. (2020) and Engel and Handel (2011) both discuss this issue, and

we now briefly cite them in the Methods. Aside from this issue, we do not have specific recoveries for our samples, which were run as single samples.

Ln 218: Was a p-value correction used to account for the number of related statistical tests performed when constructing these Corrplots? If so, the adjusted p-value should be listed here. If not, a corrected p-value should be used.

We performed a bonferroni correction to the p-values. This information was added to the methods section.

Figure 3: If possible, error bars should be added to these plots.

We added error bars to the bacterial productivity and POC concentrations. Dividing this figure into separate plots (as suggested by Reviewer #2) made this easier.

Figure 4B: If possible, error bars should be added to these plots.

We do not have sufficient replicates to calculate error bars for these plots.

Figure 5: You break up the intensity data by the extraction solvent but then do not discuss the differences between solvents. If you are able, consider combining the data into one set of plots, and move the extraction solvent data to the supplemental.

We combined the data into one set of plots, and move the extraction information to the Methods, as suggested. This suggestion was particularly helpful in leading to better data visualization, and we more extensively discuss these data in the manuscript.

Ln 310-311: This sentence is confusingly worded. Consider rewriting

We rewrote this sentence.

354: For readers who are not familiar with the hydrolysis enzymes, clarify which enzymes are endopeptidases here.

We defined the different types of enzymes/different terminology earlier in the manuscript.

Figure 9: The labels for the carbohydrate epitopes and the monosaccharides are confusing. You colored the different groups (red for epitope, orange for monosaccharide) but do not label the groups or colors anywhere. Consider either putting that in the figure text or labeling the figure with the groups (Carbohydrate epitope (red), peptidase (purple), monosaccharide (orange), etc.)

We labeled the different groups to improve readability of the figure.

437: You begin the discussion by talking about HMW biopolymers again, but that is not what the study focused on. Consider describing the formation mechanisms of POM instead.

We rewrote much of the Discussion, eliminating most of the discussion of HMW DOM.

Ln 451: Aluwihare 1997 examined HMWDOM, but this study is working with POM. Consider citing Cowie and Hedges (1984) and Sakugawa and Handa (1985) instead.

We now cite Handa and Yanagi (1969) as well as Handa and Tominaga (1969).

Ln 452: What do chl-a concentrations have to do with epitope diversity? This is an interesting result, but one that is not really discussed. Are you saying that the phytoplankton responsible for the high chl-a concentrations also produced unique carbohydrate epitopes? Or that there is a higher diversity of microbes in that sample? Your conclusion is not clear here.

We rewrote this section to clarify this point: we are suggesting that with high chl-a (indicative of a bloom/post-bloom environment), there are likely a wide variety of polysaccharides present from a range of phytoplankton, and therefore we find a higher carbohydrate epitope diversity. In this paragraph, we now cite data from spring blooms in the German Bight that have found sequential waves of phytoplankton and carbohydrate epitopes (Videl-Melgosa et al. 2021).

Ln 461: Carbohydrates and proteins are two key components (lipids also make up a substantial component) in POM, which is what was examined in the Hedges 2002 paper that is cited. They make up comparatively little of total DOM. This sentence should be updated to specify POM specifically and should say "two key components" instead of "the two key components."

**We rewrote this sentence.**

Ln 463: You say that the differences in polysaccharide hydrolase activity parallel differences in the microbial community composition, but nowhere in the results is this shown. You do not correlate enzyme activity with the abundance of specific taxa, or show concomitant shifts between enzyme activity and microbial community composition. You do show that the different stations and depths have differences in both the enzymatic activity and community composition, but no trends are discussed to show a link between those two things in this study.

It currently is not possible to connect specific organisms with specific enzyme activities, with the exception of selfish bacteria, which are not the focus here. We tried to choose wording that reflects the point that community composition and polysaccharide hydrolase activities both differ spatially. Drawing a firmer linkage that is biochemically/ecologically/biologically insightful is not yet possible; we rewrote this sentence to try to make our point more clearly.

Ln 477: This is incorrect. Although the correlations between  $\beta$ -glucosidase and the polysaccharide hydrolases had positive Pearson's r values, none of the correlations were significant. You cannot call them positive correlations if the slope of the correlation itself is not significantly different from 0. If there is a significant non-linear correlation present, additional information should be given showing that relationship and the relevant statistical tests should be performed to demonstrate its significance.

We removed this statement.

Ln 501: This sentence is difficult to follow as written. Consider rewriting to something like "While depth-stratification in microbial composition has been previously reported (e.g., DeLong et al., 2006), including in the northwestern Atlantic Ocean (Zorz et al., 2019), we found additional differences in community composition across locations (Fig. 6)."

We rewrote this sentence.

Ln 502: I am not sure why there is an "also" in this sentence. It suggests that the previous sentence was discussing differences at a different depth, but the previous sentence is discussing differences in community composition with location.

We changed 'also' to 'particularly'.

Ln 535: Cite Follett et al (2014).

We added this citation.

Ln 587: The paper title should not be capitalized

"flow" is now not capitalized, but since Upper Labrador Sea Water is a specific water mass (e.g., similar in usage to North Atlantic Deep Water), we kept the capitalization for much of the title.

Ln 723: The paper title should not be capitalized

Changed.

Ln 768: Verrucomicrobiota should be italicized

Changed.

Ln 771: The paper title should not be capitalized

Changed.

Ln 829: Zobellia galactanivorans should be italicized

Changed.

Ln 843: The paper title should not be capitalized

Changed.

**References:**

Cowie, G. L., and J. I. Hedges. 1984. "Carbohydrate sources in a coastal marine environment". *Geochimica et Cosmochimica Acta* 48: 2075–2087. https://doi.org/10.1016/0016-7037(84)90388-0.

Follett, C. L., D. J. Repeta, D. H. Rothman, L. Xu, and C. Santinelli. 2014. "Hidden cycle of dissolved organic carbon in the deep ocean". *Proceedings of the National Academy of Sciences of the United States of America* 111: 16706–16711. https://doi.org/10.1073/pnas.1407445111.

Sakugawa, H., and N. Handa. 1985. "Isolation and chemical characterization of dissolved and particulate polysaccharides in Mikawa Bay". *Geochimica et Cosmochimica Acta* 49: 1185–1193. https://doi.org/10.1016/0016-7037(85)90009-2.

The manuscript by Chad Lloyd et at. utilizes established and newer methodologies for organic matter characterization alongside bacterial community data to garner insights into mechanisms of organic matter transformation across the North Atlantic. Overall, this manuscript is well written and the organic matter methodologies and analyses are sound. I have one major concern, which is related to the 16S amplicon sequencing. The authors amplified the V3V4 hypervariable region, which is 490bp long, however they sequenced it on a HiSeq using PE250 chemistry. This means that the forward and reverse reads would only have 10 bp of overlap between them. These 10bp would also be the end bps for both reads, and typically the quality of the base pair calls near the end are quite low quality (usually less than 30), especially on the reverse read, and these base pairs are typically always trimmed off. In this case, that would not be possible. I therefore struggle to see how the forward and reverse reads could be properly aligned with such little overlap and presumably such low quality scores. I have only ever seen the V3V4 region sequenced with PE300 chemistry to substantially increase the region of overlap and allow for the trimming of terminal basepairs. As is, I am very concerned about the quality of this data, and request that the authors a) show how these forward and reverse reads can be reliably correctly aligned, b) re-sequence with different primers with greater bp overlap between forward and reverse reads, such as V1V2, V4, or V4,V5, or with PE300 chemistry, or c) remove these analyses from the manuscript.

Thank you for catching an error in our description of our primers. We made a typo when we wrote Bakt\_314F; the actual forward primer is Bakt\_34If (Herlemann et al., 2021, ISME). Combining this primer with the correctly written Bakt\_805R reverse primer would yield a V3/V4 fragment of 464 bp. With our 2x250 bp sequencing, we would be able to achieve a maximum overlap of 36 bp. While this is on the lower end of ideal overlapping lengths, our analyses (including quality trimming, merging, and chimera detection steps) produced reliable paired end sequences that reflect bacterial community compositions typical of the depths we sampled in the North Atlantic Ocean.

We note also that V3/V4 sequencing on HiSeq using PE250 chemistry has been utilized for many marine bacterial communities. For example, Fadeev et al. (2021) compared primer sets for marine microbial communities, using both Illumina MiSeq (2x300 bp) and Illumina HiSeq (2x250 bp) for various samples. Their data – irrespective of the specific sequencing technique used - demonstrated that the different primer sets yielded different observed ASVs, such that the diversity of different taxonomic groups varied at this resolution. Moreover, CARD-FISH analysis demonstrated that (for example) the gammaproteobacterial counts were lower than would be suggested by gammaproteobacterial representation in sequences. These distinctions based on different primers and different approaches to assess diversity (which have been also reported by others) highlight the point that 16s sequencing provides an inexact measure of bacterial community composition and diversity. We added Fadeev et al. (2021) and commented briefly about these points in the text.

Finally, we note that the sequencing of our samples is intended to give a broad overview of community membership at different depths and stations. We could make the same argument – that communities differ considerably by depth – by simply relying on the literature, since this observation has been made in multiple locations in the ocean. However, given that we have sequences that were derived from the same water samples from which we measured other

properties as well as and microbial enzyme activities, it is useful to include them in our manuscript.

Fadeev, Eduard, et al. "Comparison of two 16S rRNA primers (V3–V4 and V4–V5) for studies of arctic microbial communities." Frontiers in microbiology 12 (2021): 637526.

A more minor concern is that the bacterial community composition was collected on a 0.22um filter, which would typically be dominated by free-living community. These microbes are understood to primarily consume DOM, which is unfortunately not what was measured with the mono and polysaccharide composition analyses. Since particle-associated bacteria are so compositionally and metabolically distinct from their free-living counterparts, I would hesitate to draw conclusions from bulk 0.22um community data.

We now include in the text the point that the community composition analysis was intended to provide a measure of the overall composition in the depths and regions sampled, and is not intended to specifically focus on particle-associated organisms. However, we note that enzymatic activities were measured in volumes of water similar to those used for community analysis, so the spectrum of enzyme activities measured is related to the microbial community measured, in the sense that the water volumes were similar, and that the fluorescently-labeled polysaccharides are dissolved, and therefore part of the DOM pool. We also now note that in an earlier study at different stations in the western North Atlantic, we measured enzyme activities of particle-associated communities as well as (mostly free-living) communities in water volumes similar to the current study. In that study, we found that all of the activities found in the (mostly free-living) fraction were also present in the particle-associated fraction.

**Additional comments:**

Figure 2: This figure is difficult to read. If there is a way to make the stations more visually obvious, I would try to do that. I also wonder if the stations could be renamed from 17, 18, 19, and 20, to something that's a bit more intuitive for the reader, like "Shelf 1, Shelf 2, etc.

We increased the color/font contrast in Fig. 2A to improve readability. We would prefer not to rename these stations; we note that the original data, which is available on the BCO-DMO database, is linked to these specific stations under these specific names. Searching for other data from this expedition also is simplified by using the original station names. Furthermore, we also have other published manuscripts that are linked to these specific station names. For these reasons, re-naming the stations for this manuscript would likely lead to confusion.

Figure 3: This figure has a lot of data, but is very difficult to gain insight from, and I think that's largely because measurements of the same data type are not displayed next to each other. TCC is directly next to POC, Chl a, and BPP rather than other TCC values from different depths. I would recommend reorganizing so measurements of the same parameter are more in-proximity of each other

We replotted the figure, and agree that it is now easier to understand. We also added a new Suppl. Table with the actual plotted values.

Figure 4: The colors of the dots and lines are very difficult to differentiate from each other

We better differentiated the dots and lines by increasing symbol sizes.

Figure 6: Define the depth ranges for "meso" and "bottom" somewhere

We defined "meso" and "bottom" in the figure legend.

---

## Author Response (AR2)

The authors have addressed my concerns satisfactorily and substantially improved the clarity of the paper and focus of the discussion section particularly. I support its publication. There are a couple of small technical errors that need correction, but these are trivial to implement.

Ln 165: The other reviewer brought up the length of the V3/V4 fragment sequenced in the 16S rRNA analysis. The original paper listed the fragment as having a length of 490 bp. In their response to the other reviewer's comment, the authors mentioned that this was a mistake and that the actual length was 464 bp. This length has not been updated in the text, although the correct primer is now listed.

The 490 bp number refers to the overall length of the V3-V4 regions, however the primers also require some space to attach in this region, which explains the difference between overall region length and amplicon length.

Ln 376: Change "comprised" to "composed".

Changed to 'constituted'

Ln 383: Swap "prominent" and "particularly" in this sentence to improve readability.

Changed as suggested.